MiR-26a-5p inhibits GSK3β expression and promotes cardiac hypertrophy in vitro

Tang Liqun tangyz0594@163.com 1
Xie Jianhong 1
Yu Xiaoqin 2
Zheng Yangyang 1
1 Department of Geriatrics, Zhejiang Province People’s Hospital, Hangzhou Medical College , Hangzhou , Zhejiang , China
2 Department of Geriatrics, Zhejiang Aid Hospital , Hangzhou , Zhejiang , China
Fimia Gian Maria
Electronic publication date: 2020 Nov 17
Publication date: 2020
Volume: 8
Electronic Location ID: e10371
Received 2019 Oct 13; Accepted 2020 Oct 26
Copyright: ©2020 Tang et al.
Copyright year: 2020
Copyright holder: Tang et al.
License: This is an open access article distributed under the terms of the Creative Commons Attribution License, which permits unrestricted use, distribution, reproduction and adaptation in any medium and for any purpose provided that it is properly attributed. For attribution, the original author(s), title, publication source (PeerJ) and either DOI or URL of the article must be cited.
License URL: https://creativecommons.org/licenses/by/4.0/

Keywords: MiR-26a-5p, Cardiac hypertrophy, Autophagy, GSK3β, LC3, α-actinin

Funding: Zhejiang Analytical Testing Fund 2017C37068 Zhejiang Medical and Health Research Fund 2018KY259 This work was funded by the Zhejiang Analytical Testing Fund (2017C37068) and the Zhejiang Medical and Health Research Fund (2018KY259). The funders had no role in study design, data collection and analysis, decision to publish, or preparation of the manuscript.

==============================
Background

The role of miR-26a-5p expression in cardiac hypertrophy remains unclear. Herein, the effect of miR-26a-5p on cardiac hypertrophy was investigated using phenylephrine (PE)-induced cardiac hypertrophy in vitro and in a rat model of hypertension-induced hypertrophy in vivo.

Methods

The PE-induced cardiac hypertrophy models in vitro and vivo were established. To investigate the effect of miR-26a-5p activation on autophagy, the protein expression of autophagosome marker (LC3) and p62 was detected by western blot analysis. To explore the effect of miR-26a-5p activation on cardiac hypertrophy, the relative mRNA expression of cardiac hypertrophy related mark GSK3β was detected by qRT-PCR in vitro and vivo. In addition, immunofluorescence staining was used to detect cardiac hypertrophy related mark α-actinin. The cell surface area was measured by immunofluorescence staining. The direct target relationship between miR-26a-5p and GSK3β was confirmed by dual luciferase report.

Results

MiR-26a-5p was highly expressed in PE-induced cardiac hypertrophy. MiR-26a-5p promoted LC3II and decreased p62 expression in PE-induced cardiac hypertrophy in the presence or absence of lysosomal inhibitor. Furthermore, miR-26a-5p significantly inhibited GSK3β expression in vitro and in vivo. Dual luciferase report results confirmed that miR-26a-5p could directly target GSK3β. GSK3β overexpression significantly reversed the expression of cardiac hypertrophy-related markers including ANP, ACTA1 and MYH7. Immunofluorescence staining results demonstrated that miR-26a-5p promoted cardiac hypertrophy related protein α-actinin expression, and increased cell surface area in vitro and in vivo.

Conclusion

Our study revealed that miR-26a-5p promotes myocardial cell autophagy activation and cardiac hypertrophy by regulating GSK3β, which needs further research.

Introduction

Cardiac hypertrophy is a major risk factor for cardiovascular morbidity and mortality, characterized by increasing heart mass, protein synthesis, and re-expression of fetal-type genes. Cardiac overload and injury often contribute to cardiac hypertrophy. Though initially compensatory, long-term cardiac hypertrophy often induces heart failure that is a common cause of death in the world (Wang et al., 2012; Zhang et al., 2016a). Its pathological process is closely related to the complex changes of cardiac gene expression patterns and changes in molecular pathways (Tang et al., 2017). However, the regulation mechanism remains unclear (Yang et al., 2015).

As an evolutionarily conserved process, autophagy plays a key role in maintaining cardiac homeostasis. It has been confirmed that autophagy participates in regulating cardiac hypertrophy (Li et al., 2017a). Activation of autophagy is often thought to protect the heart, however, excessive autophagy induces cardiomyocyte atrophy and death (Schiattarella & Hill, 2016). Furthermore, autophagy disorders or reductions are in association with heart failure. Therefore, regulation of autophagy has become a therapeutic target for cardiovascular disease (Chen et al., 2016; Lin et al., 2016; Orogo & Gustafsson, 2015; Zhang et al., 2016b). It is critical to discover novel targets and related mechanisms to precise positioning. During autophagosome formation, the cytosolic microtubule-associated protein LC3-I is conjugated to phosphatidylethanolamine to produce LC3-II, which is recruited to the autophagosome membrane and degraded after autophagosome fusion to lysosomes. Hence, LC3-II/LC3-I ratio has been used to affect autophagy flux (Gu et al., 2016). Furthermore, Beclin-1 and p62 also have key roles in autophagy (Chen et al., 2016; Lin et al., 2016; Orogo & Gustafsson, 2015; Zhang et al., 2016b).

MicroRNAs (miRNAs) with about 22 nucleotides in length are small endogenous non-coding RNAs (Wang et al., 2012). MiRNAs can inhibit the expression of specific genes at the post-transcription level by binding to the 3′-untranslated region (3′-UTR) of the target mRNA (Wang et al., 2012). Many miRNAs have been confirmed to participate in cardiac hypertrophy (Tu et al., 2014; Verjans, Van Bilsen & Schroen, 2017; Yang et al., 2016). It has been confirmed that many miRNAs are abnormally expressed in cardiac hypertrophy, which has the effect of promoting hypertrophy or resistance to hypertrophy (Jeong et al., 2012). These miRNAs could have potential value as therapeutic targets for heart disease. Therefore, it is of importance to explore the regulation mechanisms of miRNAs in cardiac hypertrophy. For instance, overexpressed miR-146a causes cardiac hypertrophy in vivo, while mR-146a knockout attenuates hypertrophic responses and heart dysfunctions (Heggermont et al., 2017).

Growing evidence has confirmed that miR-26a-5p participates in a variety of diseases. For example, miR-26a defends vascular smooth muscle cells from H2O2-induced injury via activating PTEN/AKT /mTOR signaling axis (Peng et al., 2018). MiR-26a participates in regulating allergic inflammation (Kwon et al., 2015). Overexpressed miR-26a inhibits neuropathic pain and neuroinflammation in rat model with chronic sciatic nerve injury (Zhang et al., 2018). The roles of miR-26a in cardiovascular diseases have been pointed out in recent studies. For example, miR-26a can ameliorate the development of atherosclerosis by regulating TRPC3 (Feng, Xu & Wang, 2018). miR-26a can promote myocardial damage caused by myocardial ischemia and reperfusion (Gong et al., 2019). Up-regulation of miR-26a may promote myocardial fibrosis after acute myocardial infarction (Zhang & Cui, 2018). Yet, the function of miR-26a in cardiac hypertrophy has not been understood. In our study, we explored the regulatory function of miR-26a-5p in phenylephrine (PE)-induced cardiac hypertrophy and analyzed potential mechanisms. PE was employed to establish cardiac hypertrophy models. The results revealed that highly expressed miR-26a-5p promoted myocardial cell autophagy activation and cardiac hypertrophy in rat hearts by regulating GSK3β.

Materials and Methods

Cell culture and treatment

H9C2 cells were purchased from the Cell Bank of Sai Lan Biological Technology Co., Ltd (Zhejiang, China). The cells were maintained in Dulbecco’s Modified Eagle Medium (Invitrogen, Carlsbad, CA, USA) supplemented with 10% fetal bovine serum (Invitrogen, Carlsbad, CA, USA) in a humidified atmosphere (37 °C, 5% CO2). The cells were treated by different concentrations of PE (10  µM; 20 µM; 50 µM; 100 µM; 200 µM) for 48 h. MiR-26a-5p agomir and inhibitor were used to transiently transfect H9C2 cells using Lipofectamine 2000. Lysosomal inhibitor Bafilomycin A1 (Baf-A1; Sangon Biotech; Shanghai, China) was dissolved to 100 µg/ml stock solution and diluted to 10 nM to inhibit autophagy of H9C2 cells.

Transfection

MiR-26a-5p agomir, miR-26a-5p inhibitor and corresponding controls were purchased from Genepharma (Shanghai, China). GSK3β full length (forward, 5′-GAAGATTCTAGAGCTAGCGAATTCGCCACCATGTCGGGGCGACCGAGAAC-3′; reverse, 5′-GCGGCCGCGGATCCTCAGGTAGAGTTGGAGGCTGATG-3′) and GSK3β siRNAs (siRNA1, 5′-GGAGAGCCCAATGTTTCAT3′; siRNA2, 5′-CCGATTACACGTCTAGTAT-3′; siRNA3, 5′-GCTGTGTGTTGGCTGAATT-3′) were designed and synthesized by General biosystems (Anhui, China). Transient transfection into H9C2 cells was performed using the Lipofectamine 2000.

Quantitative real-time polymerase chain reaction (qRT-PCR)

Total RNA was extracted from H9C2 cells or tissues using TRIzol reagent (Takara, Beijing, China). After that, the extracted RNA was reverse transcribed into cDNA using a RevertAid First Strand cDNA synthesis Kit (Thermo, Waltham, Massachusetts, USA). SYBR Green Master Mix (Takara, Beijing, China) was used for qRT-PCR. The PCR primer information of miR-26a-5p, U6, glycogen synthase kinase 3 β (GSK3β), ANP, ACTA1, MYH7 and GAPDH is shown in Table 1. U6 or GAPDH served as internal miRNA controls. The relative expression of target genes was determined by 2−ΔΔCt method.

Table 1 The sequence of primers for qRT-PCR.

Target	Sequence (5′–3′)	
Rat miR-26a-5p	5′-ACACTCCAGCTGGGTTCAAGTAATCCAGGA -3′(forward)
5′-CTCAACTGGTGTCGTGGAGTCGGCAATTCAGTTGAGAGCCTATC-3′(reverse)	
Rat U6	5′-CTCGCTTCGGCAGCACA-3′(forward)
5′-AACGCTTCACGAATTTGCGT-3′(reverse)	
Rat GSK3 β	5′-AAGCCCAGCCTACTAACAACC-3′(forward)
5′-CAGCCCCACTGTACTGACTG-3′(reverse)	
Rat ANP	5′-GGGCTTCTTCCTCTTCCTG-3′(forward)
5′-CGCTTCATCGGTCTGCTC-3′(reverse)	
Rat ACTA1	5′-CTCTTGTGTGTGACAACGGC-3′(forward)
5′-CCCATACCGACCATGACACC-3′(reverse)	
Rat MYH7	5′-TCAGTCATGGCGGATCGAG-3′(forward)
5′-ACAGTCACCGTCTTGCCATT-3′(reverse)	
Rat GAPDH	5′-GGAGCGAGATCCCTCCAAAAT-3′(forward)
5′-GGCTGTTGTCATACTTCTCATGG-3′(reverse)	
Rat GSK3β	5′-GAGACACACCTGCCCTCTTC-3′(forward)
5′-TGGGGCTGTTCAGGTAGAGT-3′(reverse)	
Rat GAPDH	5′-GCAAGTTCAACGGCACAG-3′(forward)
5′-GCCAGTAGACTCCACGACAT-3′(reverse)	

Western blot analysis

H9C2 cells pretreated were divided into 6 groups including PE, miR-26a agomir, miR-26a agomir and PE, miR-26a inhibitor, miR-26a inhibitor and PE, and control group. H9C2 cells were lysed using RIPA buffer with a protease inhibitor cocktail (Sigma, NY, USA), which was separated using SDS-PAGE. After that, the protein was transferred onto PVDF membrane (Millipore, NY, USA), and blocked with 5% skim milk solution for 1 h. The membrane was incubated with primary antibodies including anti-LC3 (1:1000, 12741S, CST, Danvers, MA,USA) , anti-Beclin-1 (1:1000, CST), anti-p62 (1:1000, CST), anti-α-actinin (1:200, abcam, UK), anti-GSK3β (1:1000, CST) and anti-GAPDH (1:5000, 2118L, CST) overnight at 4 °C and then incubated with secondary antibody for 1 h at room temperature. Finally, ECL HRP Substrate Kit (Bio-Rad, Hercules, CA) was used to visualize the proteins. GAPDH served as an internal control.

Immunofluorescence assay

Treated H9C2 cells were climbed in 24-well plates. On the second day, cells were transfected with miR-26a-5p agomir, miR-26a-5p inhibitor or the corresponding controls and treated with PE for 48 h for immunofluorescence staining. Then, the medium was removed and rinsed 3 times with 1 × PBS. The cells were fixed with 4% paraformaldehyde at room temperature for 20 min, and then 0.1% Triton X-100 solution was added at room temperature for 10 min. The cells were blocked with 3% BSA solution at 37 °C for 90 min. After that, the cells were incubated with primary antibody including anti- α-actinin (1:200, abcam, UK) or anti-LC3 (1:1000, CST, Danvers, MA, USA) at 4 °C overnight, followed by incubation with fluorescent secondary antibody (1: 2000) at 37 °C for 1.5 h in the dark. Then, the cells were stained with 4′, 6-diamidino-2-phenylindole (DAPI) for 5 min at room temperature. After mounting, the cells were observed under a fluorescent microscope (ECLIPSE Ti-S; Nikon, Tokyo, Japan).

Measurement of cell surface areas

After climbing, the cells were fixed with 4% paraformaldehyde for 15 min, followed by permeation using 0.5% Triton X-100 at room temperature for 20 min. Then, the cells were incubated with TRITC-phalloidin antibody (1:200; Shanghai Yisheng Biotechnology Co., Ltd., Shanghai, China) at room temperature in the dark for 30 min. The cells were then incubated with DAPI in the dark for 5 min. After washing away the excess DAPI using PBS, the liquid on the slide was absorbed with absorbent paper. The slide was mounted with the mounting solution containing anti-fluorescence quencher, and the images were observed under a fluorescent microscope.

Luciferase reporter assay

The 3′-UTR region of GSK3β containing amplified sites (wide type, Wt) and mutant sites (mutant type, Mut) were conducted and cloned into pGL3-CM vectors (Ambion, Grand Island, NY, USA). Then, plasmids were co-transfected into H9C2 cells with miR-26a-5p agomir or NC using lipofectamine 2000 reagent (life technologies, Carlsbad, California, USA). Then, firefly luciferase activities were measured using Dual-Luciferase reporter assay system (Promega, Madison, WI, USA). Relative luciferase activity was normalized to Renilla luciferase activity.

Animal experiments

Male spontaneously hypertensive rats (SHR; 12-weeks old; weighing 200–250 g) were purchased from Shanghai Experimental Animal Center (Shanghai, China), which were used to construct a compensatory cardiac hypertrophy model. All rats were reared under 12/12 cycle of light at room temperature (25−27 °C) and fed with a regular diet. They were randomly divided into two groups: control (n = 4), NC agomir group (n = 4) and miR-26a-5p agomir group (n = 10). All rats were injected with miR-26a-5p agomir or NC agomir (10 mg/kg) in the tail vein. After continuous administration for 3 days, a total of 5 times, all rats were sacrificed with an overdose of 2% sodium pentobarbital. Fresh heart tissues were removed for further analysis. All animals were treated in accordance with the Guide for the Care and Use of Laboratory Animals. Our study was approved by the Ethics Committee of Zhejiang Province People’s Hospital (20190232).

Hematoxylin and eosin (H&E)

After harvesting and separating the hearts, the fresh heart tissues were fixed in 10% formalin solution for 24–48 h, and then dehydrated by 70%, 80%, 90%, 95%, 100% ethanol I, and 100% ethanol II, for 1 h each time. Next, the tissues were treated with xylene I for 10 min, xylene II for 20 min, and then embedded in paraffin. The tissues were cut into 4 µM thickness, and then placed in a 65 °C box for 6–12 h. H&E staining was presented. Briefly, after the sections were dewaxed to water, the sections were immersed in hematoxylin staining solution at room temperature for 10 min and eosin staining solution for 5 s. After dehydration, transparency, and sealing, the pathological changes of myocardial tissue were observed under a microscope (200 ×).

Statistical analysis

Statistical analysis was carried out using GraphPad Prism 7.0 and SPSS 23. Results were presented as the mean ± standard deviation, except for special. Differences between two groups were assessed by Student’s t test, while two-way analysis of variance (ANOVA) followed by Tukey post hoc testing was used for multiple comparisons. P-value <  0.05 was considered statistically significant.

Results

MiR-26a-5p is highly expressed in PE-induced cardiac hypertrophy

Firstly, we assessed whether the cardiac hypertrophy model was successfully built in vitro. H9C2 cells were treated with different concentrations of PE in vitro. qRT-PCR results showed that the expression of miR-26a-5p in H9C2 cells was significantly elevated in PE-induced cardiac hypertrophy, with a dose-dependent manner (Fig. 1A). After treatment with 200 µM PE, miR-26a-5p had the highest expression level in H9C2 cells. Therefore, 200 µM PE was used to conduct cardiac hypertrophy models.

Figure 1 MiR-26a promotes myocardial cell autophagy activation in PE-induced cardiac hypertrophy.

(A) The mRNA expression levels of miR-26a-5p in H9C2 cells treated by different concentrations of PE. (B–E) Western blot analysis results showing the expression of LC3II, Beclin-1 and p62. (F–H) The expression of LC3II and p62 in H9C2 cells in the presence or absence of Baf-A1. Data were presented as mean ±  SD (n = 3). ∗p < 0.05; ∗∗p < 0.01; ∗∗∗∗p < 0.0001.

MiR-26a-5p promotes myocardial cell autophagy activation in PE-induced cardiac hypertrophy

Increasing evidence suggests that cardiomyocyte autophagy is closely related to cardiac hypertrophy. Thus, autophagy-related markers including LC3 (LC3-II and LC3-I), Beclin-1 and p62 were detected in H9C2 cells. We found that LC3II and Beclin-1 protein expression was markedly increased in PE-induced cardiac hypertrophy, while p62 protein expression was significantly decreased (Figs. 1B, 1C). Intriguingly, miR-26a-5p agomir dramatically increased LC3II and Beclin-1 expression and decreased p62 expression in PE-induced H9C2 cells (Figs. 1B–1E). Instead, miR-26a-5p inhibitor significantly reversed PE-induced changes in Beclin-1 and p62 proteins (Fig. 1D, 1E). In the presence or absence of lysosomal inhibitor Baf-A1, we compared LC3II and p62 expression levels in H9C2 cells. The results showed that miR-26a-5p agomir distinctly increased LC3II and decreased p62 expression in PE-induced H9C2 cells in the presence or absence of Baf-A1 (Figs. 1F–1H). Similar results were observed under immunofluorescence. miR-26a-5p agomir remarkably enhanced PE-induced increase in LC3II expression, on the contrary, miR-26a-5p inhibitor reversed PE-induced increase in LC3II expression (Figs. 2A–2S). Above results suggested that autophagy was activated in PE-induced cardiac hypertrophy.

Figure 2 Immunofluorescence results showing the effect of miR-26a-5p on LC3 protein expression in PE-induced cardiac hypertrophy.

(A–R) Representative images of immunofluorescence results (×200). (S) The expression of LC3 protein was measured in H9C2 cells treated with 200 µM PE. Data were presented as mean ±  SD (n = 3). ∗p < 0.05; ∗∗∗∗p < 0.0001.

MiR-26a-5p inhibits GSK3β expression and promotes cardiac hypertrophy in vitro

To explore miR-26a-5p effects on cardiac hypertrophy, biomarker proteins related-to cardiac hypertrophy were tested. Firstly, we would confirm whether miR-26a-5p participated in regulating cardiac hypertrophy. qRT-PCR results revealed that the expression of miR-26a-5p in PE-induced cardiac hypertrophy was remarkably up-regulated compared with control group (Fig. 3A). As expected, miR-26a-5p agomir significantly increased its expression in H9C2 cells treated with 200  µM PE. Meanwhile, the expression of miR-26a-5p was remarkedly decreased in PE-induced H9C2 cells (Fig. 3A). To further assess the function of miR-26a-5p in cardiac hypertrophy, we tested cardiac hypertrophy related marker protein GSK3β (a negative regulator of cardiac hypertrophy) expression. At the mRNA and protein levels, the expression of GSK3β was decreased in PE-induced cardiac hypertrophy compared to control group (Figs. 3B–3D). However, miR-26a-5p inhibitor significantly elevated the expression of GSK3β in H9C2 cells treated with PE (Figs. 3B–3D). Dual luciferase report results confirmed the direct target relationship between miR-26a-5p and GSK3β 3′UTR. miR-26a-5p overexpression could reduce luciferase activity of wild type GSK3β, however such effects were not investigated for mutation type GSK3β (Figs. 3E, 3F). According to above results, miR-26a-5p inhibited antihypertrophic GSK3β expression in PE-induced cardiac hypertrophy in vitro.

Figure 3 MiR-26a-5p inhibits GSK3β expression and promotes cardiac hypertrophy in vitro.

(A, B) qRT-PCR results showing the expression levels of miR-26a-5p and GSK3β in H9C2 cells treated with PE. (C, D) Western blot showing the effect of miR-26a-5p on the expression of GSK3 β and α-actinin proteins. (E, F) Dual luciferase report results confirmed that miR-26a-5p could bind to GSK3β 3′ UTR. (G) The effect of α-actinin on the mRNA expression of α-actinin according to qRT-PCR results. (H–Z) Immunofluorescence results showing the effect of miR-26a-5p on α-actinin protein in PE-induced cardiac hypertrophy (×200). The marker of α-actinin was stained by green color. Data were presented as mean ±  SD (n = 3). ∗p < 0.05; ∗∗p < 0.01; ∗∗∗p < 0.001; ∗∗∗∗p < 0.0001.

The expression of cardiac hypertrophy-related protein α-actinin was also examined in treated H9C2 cells. As shown in qRT-PCR results, the mRNA expression level of α-actinin was remarkedly elevated in H9C2 cells after treatment with PE (Fig. 3G). Also, PE-induced increase in α-actinin expression was significantly strengthened by miR-26a-5p agomir, and was significantly decreased by miR-26a-5p inhibitor (Fig. 3G). Similar results were investigated in immunofluorescence staining results (Figs. 3H–3Z). To further investigate the effect of miR-26a-5p on cardiomyocyte surface areas, we performed TRITC-phalloidin fluorescence staining. Compared to control group, PE remarkedly promoted cardiomyocyte surface areas (Figs. 4A–4S). As expected, miR-26a-5p agomir significantly increased PE-induced cardiomyocyte surface areas (Figs. 4A–4S). Instead, miR-26a-5p inhibitor reversed PE-induced cardiomyocyte surface areas (Figs. 4A–4S). These results revealed that miR-26a-5p promoted cardiac hypertrophy in vitro.

Figure 4 Effects of miR-26a-5p on the surface areas in PE-induced cardiac hypertrophy.

(A–R) Representative images of TRITC-phalloidin fluorescence staining. (S) Relative cardiomyocyte surface areas. Data were presented as mean ±  SD (n = 3). ∗p < 0.05; ∗∗p < 0.01; ∗∗∗∗p < 0.0001.

miR-26a-5p promotes cardiac hypertrophy by regulating GSK3β

We further observed the effects of miR-26a-5p on cardiac hypertrophy-related markers including ANP, ACTA1 and MYH7. qRT-PCR results showed that the mRNA expression levels of ANP, ACTA1 and MYH7PE were significantly elevated in PE-induced cardiomyocytes (Figs. 5A–5C). PE-induced increase in cardiac hypertrophy-related markers was dramatically promoted by miR-26a-5p agomir and was reversed by its inhibitor (Figs. 5A–5C). To confirm that miR-26a-5p could promote cardiac hypertrophy by regulating GSK3 β, GSK3β was successfully overexpressed and silenced according to qRT-PCR results (Figs. 5D, 5E). As shown in Figs. 5F–5H, GSK3β overexpression remarkedly decreased miR-26a-5p-induced increase in cardiac hypertrophy-related marker expression. Furthermore, silencing GSK3β significantly reversed miR-26a-5p inhibitor-induced decrease in cardiac hypertrophy-related marker expression in PE-induced H9C2 cells. Above results indicated that miR-26a-5p could promote cardiac hypertrophy by regulating GSK3β.

Figure 5 miR-26a-5p promotes cardiac hypertrophy by regulating GSK3 β.

(A–C) qRT-PCR results showing the effects of miR-26a-5p on PE-induced H9C2 cells. (D, E) GSK3β was successfully overexpressed and silenced in H9C2 cells. (F–H) According to qRT-PCR results, miR-26a-5p could promote the expression levels of cardiac hypertrophy-related markers by regulating GSK3β. Data were presented as mean ±  SD (n = 3). ∗∗p < 0.01; ∗∗∗p < 0.001; ∗∗∗∗p < 0.0001.

MiR-26a-5p inhibits GSK3β expression and promotes cardiac hypertrophy in vivo

To confirm whether overexpressed miR-26a-5p promoted cardiac hypertrophy, miR-26a-5p agomir or NC agomir was injected into SHR. The results showed that miR-26a-5p expression was elevated in SHR injected miR-26a-5p agomir compared to control groups (Fig. 6A). Additionally, we found that GSK3β mRNA and protein expression was both down-regulated in SHR hearts after injecting miR-26a-5p agomir compared to controls (Figs. 6B, 6C). Furthermore, our western blot results showed that compared to control agomir group, LC3II and Beclin-1 expression was significantly elevated and p62 expression was significantly decreased in miR-26a-5p agomir group (Figs. 6D–6G). In Figs. 6H–6P, the cell surface area was increased in miR-26a-5p agomir group compared with control agomir group. qRT-PCR showed that cardiac hypertrophy-related markers including ANP, ACTA1 and MYH7 were significantly elevated in miR-26a-5p agomir group compared to controls (Figs. 6Q–6S). Therefore, above results suggested that miR-26a-5p inhibited GSK3β expression and promoted cardiac hypertrophy in vivo.

Figure 6 MiR-26a-5p inhibits GSK3 β expression and promotes cardiac hypertrophy in vivo.

(A, B) qRT-PCR showing the mRNA expression levels of miR-26a-5p and GSK3β in SHR injected with miR-26a agomir. (C–G) The protein expression levels of GSK3β, LC3II, Beclin-1 and p62 were detected in myocardial tissues by western blot. (H–P) Hematoxylin and eosin (H&E) staining of myocardial tissues in the three groups (×200). (Q–S) qRT-PCR results showing the mRNA expression levels of cardiac hypertrophy-related markers including ANP, ACTA1 and MYH7 in myocardial tissues. Data were presented as mean ±  SD n = 3). ∗∗p-value < 0.01; ∗∗∗∗p-value <0.0001; ns, no statistical significance.

Discussion

It has been confirmed that pathological hypertrophy may result in increased interstitial fibrosis, cell death and cardiac dysfunction (Dong et al., 2018). As a hallmark of cardiac ageing, cardiac hypertrophy may induce an increased incidence of cardiac disease (Woodall & Gustafsson, 2018), which maintains cardiac homeostasis in a physiological environment (Bernardo et al., 2010). Due to the limited ability of adult myocardium to regenerate, the loss of functional cardiomyocytes caused by autophagy is one of the potential mechanisms of myocardial remodeling and cardiac disease (Sun et al., 2018). In our study, miR-26a-5p was up-regulated in PE-induced cardiac hypertrophy, furthermore, high miR-26a expression promoted myocardial cell autophagy activation and cardiac hypertrophy by GSK3β (Gu et al., 2016).

As an endogenous regulator, miRNA participates in mediating post-transcriptional gene expression (Sadiq et al., 2017), while cardiac hypertrophy is largely dependent on gene expression (Qi et al., 2019). Therefore, there is reason to believe that miRNAs participate in cardiac hypertrophy (Gupta & Thum, 2016). For example, miR-199a inhibits autophagy and contributes to cardiac hypertrophy (Li et al., 2017b). MiR-181a has been confirmed to be highly expressed in Ang II-induced cardiac hypertrophy and mediate autophagy (Li, Lv & Gao, 2017). MiR-365 induces cardiac hypertrophy and inhibits autophagy through mediating Skp2 expression (Wu et al., 2017). MiR-208a-3p activates autophagy in Ang II-induced cardiac hypertrophy via PDCD4-ATG5 pathway (Wang et al., 2018). MiR-22 also has been found to be a key regulatory factor of cardiac autophagy (Gupta et al., 2016). Yet, the function of miR-26a-5p in cardiac hypertrophy remains unclear. In our study, we successfully constructed cardiac hypertrophy models in vitro and in vivo. MiR-26a-5p was up-regulated in PE-induced cardiac hypertrophy. It has been found that LC3, Beclin-1 and p62 are key regulators of autophagy, and reversion of their expression could attenuate cardiac hypertrophy (Noh et al., 2016; Pan et al., 2017; Xie et al., 2018). Therefore, the expression levels of LC3, Beclin-1 and p62 can reflect the autophagy in cardiac hypertrophy. In our study, after PE-induced cardiac hypertrophy cells were treated by miR-26a-5p agomir, the expression of LC3II and Beclin-1 proteins was increased and p62 protein expression was decreased in the presence or absence of lysosomal inhibitor, suggesting that high miR-26a-5p expression may promote autophagy in cardiac hypertrophy.

GSK3β regulates various cellular functions by phosphorylating cellular substrates (Hardt & Sadoshima, 2002). The activation of GSK-3β is inhibited through PKB/Akt and Wnt signaling pathways. Growing studies suggest that GSK-3β negatively regulates cardiac hypertrophy, and inhibits GSK3β via hypertrophic stimulation contributes to cardiac hypertrophy. Because of its phosphorylatio, inhibition of GSK3β is associated with cardiac hypertrophy in response to endothelin-1 or phenylephrine (Li et al., 2007; Markou et al., 2008). Our results revealed that GSK3β expression was inhibited in PE-induced cardiac hypertrophy treated by miR-26a agomir, which was consistent with previous studies. More importantly, dual luciferase report results confirmed that miR-26a-5p could bind to GSK3β 3′UTR. miR-26a-5p dramatically promoted PE-induced increase in cardiac hypertrophy-related markers including ANP, ACTA1 and MYH7. GSK3β could remarkedly reversed miR-26a-5p-induced increase in cardiac hypertrophy-related marker expression. Thus, our findings revealed that miR-26a-5p could promote cardiac hypertrophy by regulating GSK3β.

It has been confirmed that α-actinin is up-regulated in cardiac hypertrophy, which has been a hallmark of cardiac hypertrophy. Angiotensin II promotes α-actinin expression in cardiac fibroblasts (Kawano et al., 2000; Sheng et al., 2016). In our study, qRT-PCR and immunofluorescence staining results demonstrated that α-actinin expression was increased in PE-induced cardiac hypertrophy transfected by miR-26a-5p agomir. Therefore, overexpressed miR-26-5p induced cardiac hypertrophy by promoting α-actinin expression. To further confirm whether overexpression of miR-26a-5p could contribute to cardiac hypertrophy, we measured cell surface area. We found that cell surface area was significantly increased in H9C2 cells transfected by miR-26a-5p agomir, which confirmed our conclusion.

In addition, we successfully established cardiac hypertrophy models in vivo. qRT-PCR analysis suggested that GSK3β expression was decreased in SHR transfected by miR-26a agomir. As shown in H&E staining results, overexpressed miR-26a-5p could aggravate cardiac hypertrophy. Furthermore, miR-26a-5p agomir significantly promoted cardiac hypertrophy-related markers including ANP, ACTA1 and MYH7. Therefore, high miR-26a-5p expression inhibited GSK3β expression and promoted cardiac hypertrophy in vivo.

Conclusion

Our findings reveal that miR-26a-5p promotes PE-induced cardiac hypertrophy by regulating GSK3β and activates autophagy in vitro and in vivo, which may provide a novel insight into the pathogenesis of cardiac hypertrophy.

Supplemental Information

Supplemental Information 1 Raw data

Click here for additional data file.

Abbreviations

miRNAs microRNAs

PE phenylephrine

qRT-PCR quantitative real-time polymerase chain reaction

SHR spontaneously Hypertensive Rats

H&E Haematoxylin-Eosin

GSK3β glycogen synthase kinase 3β

DAPI 4′, 6-diamidino-2-phenylindole.

Additional Information and Declarations

Competing Interests

Author Contributions

Animal Ethics

Data Availability

The authors declare there are no competing interests.

Liqun Tang conceived and designed the experiments, performed the experiments, authored or reviewed drafts of the paper, and approved the final draft.

Jianhong Xie performed the experiments, prepared figures and/or tables, authored or reviewed drafts of the paper, and approved the final draft.

Xiaoqin Yu performed the experiments, analyzed the data, prepared figures and/or tables, and approved the final draft.

Yangyang Zheng analyzed the data, prepared figures and/or tables, and approved the final draft.

The following information was supplied relating to ethical approvals (i.e., approving body and any reference numbers):

The study was approved by the Ethics Committee of Zhejiang Province People’s Hospital (20190232).

The following information was supplied regarding data availability:

The raw measurements are provided in the Supplementary Files.

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
