# Peer review of "MiR-26a-5p inhibits GSK3β expression and promotes cardiac hypertrophy in vitro"

_PeerJ, doi:10.7717/peerj.10371_

## Round 0.1 · original submission · Major Revisions

All referees considered your work of interest; however they raised several important concerns that need to be addressed to support the main conclusions of the manuscript. In particular, they requested a more detailed analysis of autophagy levels both in vitro and in vivo, and a better characterization of the relationship between miR-26a-5p-mediated decrease of GSK3β levels, autophagy modulation and PE-induced cardiac hypertrophy.

Reviewer 1 ·

Basic reporting

The article includes adequate introduction and background to evaluate the work into the specific context of interest.

Experimental design

Methods were described with sufficient details. Supplementary data should be better described.

Validity of the findings

More experimental results are required to elucidate the correlation between autophagy and cardiac hypertrophy.

Additional comments

In this manuscript, Liqun Tang et al. propose to investigate the role of miR-26a-5p expression in the regulation of autophagy and cardiac hypertrophy. This paper contains some interesting observations about the role of miR-26a-5p in cardiac hypertrophy, however there are a consistent number of concerns that need to be addressed by the authors to adequately validate these observations.
Major concerns:
1) The correlation between the induction of autophagy by miR-26a-5p upon phenylephrine (PE) treatment and its ability to promote cardiac hypertrophy by reducing GSK3β expression is not discussed by the authors and further work is needed to better characterize the role of autophagy induced by miR-26a-5p upon phenylephrine (PE)-induced cardiac hypertrophy. It would be important to provide additional experimental evidences in order to assess whether the up-regulation of autophagy by miR-26a is effectively responsible for the cardiac hypertrophy stimulation. To this aim, the authors should analyze if the decreased expression of GSK3β observed during (PE)-induced cardiac hypertrophy in presence of miR-26a is affected by treating cells with the autophagy inhibitor 3-methyladenine or by silencing specific autophagic genes, such as Beclin1 or ATGs. (Fig. 2)

2) The authors analysed the levels of lipidated LC3 after PE treatment in combination with miR-26a agomir or miR-26a inhibitor. In Fig. 1B it is shown that LC3-II/LC3-I ratio (please do not indicated this as LC3b/LC3a because these names refer to different LC3 genes) in PE-induced cardiac hypertrophy treated with miR-26a agomir group was higher than control groups. However, according to the published guidelines for monitoring autophagy (Klionsky DJ et al., 2016), the authors should analyse LC3 lipidation by measuring LC3II/GAPDH ratio. In addition, according to the published guidelines for monitoring autophagy (Klionsky DJ et al., 2016), authors should analyse autophagy flux by comparing LC3 levels in the presence or absence of lysosomal inhibitors to rule out that the increase of LC3-II levels is due to a block in autophagosome-lysosome fusion. Moreover, authors should analyse autophagic cargo degradation by analyzing p62 protein levels, in the presence or absence of lysosomal inhibitors. (Fig. 1)

3) The authors confirmed by in vivo experiments the ability of miR-26a to inhibit GSK3β expression and promote cardiac hypertrophy, but they did not perform any in vivo experiments on the autophagy regulation by miR-26a. To this aim, authors should analyse LC3 and p62 protein levels in this in vivo system. (Fig. 4)

4) The authors showed that GSK3β expression is inhibited by miR-26a at transcriptional level. To validate this finding, authors should analyse also GSK3β protein levels in H9C2 cells and in SHR rats. (Fig. 2-4)

Minor concerns:
1) All the supplementary data should be better organized and classified. In particular, the authors should provide more details on the supplementary tables where quantification data are present. It is not clear to which figures they refer and the meaning of the numeric values.
2) Figure 2 title should be corrected in “MiR-26a inhibits GSK3β expression and promotes cardiac hypertrophy in vitro”

3) The abbreviations should be explained with the full name the first time they appear in the main text:
- PE line 75
- CK line 134
- SHR line 179

4) Line 163: mark should be corrected with marker

5) Line 138 - 140 – 183 – 238: obviously should be removed

Reviewer 2 ·

Basic reporting

(1) Page 8 line 134: Please add the full name of short-form “CK”.
(2) Page 8 line 144-145: Through cell experiments, the authors concluded that "miR-26a promoted myocardial cell autophagy activation in PE-induced cardiac hypertrophy." However, as shown in Figure 1B-E, I find no significant changes of LC3b/LC3a levels in H9C2 cells transfected with miRNA-26a inhibitor after PE stimulation compared to cells stimulated with PE alone. In my opinion, this means that reducing miRNA-26a expression in cardiac hypertrophy has no effect on autophagy activation. I feel sorry that I cannot understand the statement, and I am worried that the figure may contradict the conclusion authors have drawn in this section. Please check the results carefully and, when necessary, supplement relevant description and analysis in this part.
(3) Page 16 Figure 1C-E: Please add error bars to the histograms and clearly mark the compared groups and P values.
(4) Page 17-18 Figure 2C: In figure legends, the authors stated that “the cardiac hypertrophy marker protein α-actinin (red)”. However, the actual immunofluorescence staining of α-actinin was green. Please correct this mistake.
(5) Figure 2C, Figure 3A, and Figure 4C: Please add appropriate scale bars to these figures.
(6) Page 19 Figure 3A: I couldn’t see what is in this figure. Please pay attention to this issue and provide clearer images.
(7) Page 21 Figure 4C: I find it difficult to clearly distinguish the difference of cardiomyocyte size between these two groups. Please provide a higher magnification image or quantitative analysis of cardiomyocyte size.
(8) Page 23 Table 1: Please explain why the listed sequence of primers for qRT-PCR contained homo primers such as Homo U6, Homo GSK3β, and Homo GAPDH since the experimental subjects were rat cardiomyocyte cell line (H9c2 cells) and rats.

Experimental design

(1) Page 7 line 114-116:It could be better to describe the animal experiments in more detail, including the age of the spontaneously hypertensive rats, the husbandry condition, and whether it complied with the animal experiment ethics. Furthermore, the results section mentioned the observation of PE-induced cardiac hypertrophy (Page 9 line 183), but the methods section did not mention the intervention of PE in rats. Please revise to ensure the uniformity of the experiment.
(2) Page 8 line 135-145: The primary antibody anti-LC3 (1:1000, 12741S, CST) involved in this study was used to recognize endogenous levels of total LC3A and LC3B proteins according to the manufacturer's instructions. In Western blot analysis, two bands with molecular weights of 16 kDa and 14 KDa appeared, which represent LC3A/B-I and LC3A/B-II, respectively. I couldn’t understand why the authors analyzed relative LC3a level, LC3b level, and LC3b/LC3a instead of LC3A/B-II/LC3A/B-I to observe autophagy levels.
In addition, it could be better to supplement the protein level of p62/SQSTM1, which is another commonly used autophagic marker.
(3) Page 8 line 152 and Page 9 line 177: The authors observed that MiR-26a inhibits GSK3β expression in cardiac hypertrophy, both in vivo and in vitro. So, does miR-26a directly target GSK3β? It could be better to explore and verify the downstream target genes of miR-26a.

Validity of the findings

Page 6 line 69-74:This section described the research background associated with miR-26a, including its protection of vascular smooth muscle cells from H2O2-induced injury, mediating allergic inflammation, and inhibition of neuropathic pain and neuroinflammation. However, I think these previous studies are not enough to be the reason for choosing miR-26a. Please investigate and supplement the progress of research on the role of miR-26a in cardiovascular diseases.

Additional comments

This is a good original primary research which sought to explore the role of miR-26a-5p in cardiac hypertrophy and its specific mechanisms. After detecting the high expression of miR-26a-5p in PE-induced hypertrophic cardiomyocytes, the authors further explored the mechanism by using miR-26a agomir and inhibitor to transfect H9c2 cells. Finally, miR-26a-5p was found to significantly inhibit GSK3β expression in vitro and in vivo, and also promote the autophagy activation of cardiomyocytes. The topic of this manuscript is timely and interesting, which provides new insights into the pathogenesis of cardiac hypertrophy. However, substantial efforts will be needed to improve the logic of the article, experimental design, and professionalism of figures. In conclusion, this manuscript cannot be accepted in the current situation. Perhaps, after major revisions, it can be accepted.

Reviewer 3 ·

Basic reporting

No comment

Experimental design

No comment

Validity of the findings

No comment

Additional comments

In this manuscript Tang and colleagues studied the effects of miR-26a on cardiac hypertrophy in both in vitro, phenylephrine (PE) treated H9C2 cells, and in vivo, spontaneously hypertensive rats (SHR) models. They found that miR-26a-5p was highly expressed in PE-induced cardiac hypertrophy as well as the LC3 levels. Moreover, miR-26a agomir induced LC3 accumulation and the cardiac hypertrophy related protein α-actinin expression, while inhibiting the expression of GSK3β, a negative regulator of cardiac hypertrophy. Finally, the PE and the miR-26a agomir treatments increased the cellular hypertrophy both in vitro and in vivo.

Although the work is potentially interesting one of the main conclusion drawn by the authors (i.e. miR-26a-5p activates myocardial cell autophagy) is not supported by the data presented.

In fact, the authors stated:

- Line 138. “protein expression levels of LC3-a and LC3-b were obviously elevated in H9C2 cells treated by 200 μM PE, suggesting that the autophagy was activated in PE-induced cardiac hypertrophy compared to controls”.
- Line 144. “Therefore, miR-26a promoted myocardial cell autophagy activation in PE-induced cardiac hypertrophy”.

The authors reported increased levels of LC3 protein upon PE treatment (+/- miR-26a agomir); this is not sufficient to state that these treatments induced the autophagic flux, in fact an increased LC3 levels could be due to either an increased autophagosomes formation or an inhibition of autophagosomes degradation. LC3 levels should be measured in presence of an autophagy inhibitor (NH4Cl; Bafilomycin etc.; See Rubinsztein DC et al. Autophagy. 2009 Jul;5(5):585-9).

Other main points:

-ABSTRACT. The authors stated “MiR-26a-5p was highly expressed in PE-induced cardiac hypertrophy. MiR-26a-5p promoted LC3b/LC3a ratio in PE-induced cardiac hypertrophy”.
Generally, the authors confused LC3-a and LC3-b (genes) with LC3-I and LC3-II (proteins) forms. Authors showed results only about LC3-I and LC3-II forms.

- Line 129. The authors stated “Firstly, we assessed whether the cardiac hypertrophy model was successfully built in vitro”. However, no data were presented about the suitability of the hypertrophy cellular model. Moreover, in figure 3B it seems that the PE treatment does not induce cellular hypertrophy.


-Figures 1C, 1D and 1E. How many experiments were performed? Standard deviations and statistic should be performed.

-The quality of figure 3A is too poor.

-Figure 4. The enlargement of cells of miRNA-26 agomir treated SHR rats is not clear, an enlargement of the picture is required.

-Figure 4. It would be very interesting to compare the levels of GSK-3 and miR-26a of SHR to those of healthy rats.

-Figure 4. It would be very interesting to measure LC3 levels in the SHR rats, treated with either miR26 agomir or NC agomir, and comparing them to those of healthy rats..

-Figure 4. It would be important to show, as a control, the levels of known miRNA-26 target genes, in both H9C2 cells and SHR rats’ heart

- In the Discussion section would be important for the readers to have hypothesis on how miR-26a can affect GSK3 and α-actinin.

-Text should be checked carefully by a native english speaker

Minor points

- Line 157. The author stated “To further assess the function of miR-26a in cardiac hypertrophy, we tested cardiac hypertrophy related mark protein GSK3β expression”. It should be stated clearly that GSK-3β is a negative regulator of cardiac hypertrophy.

- Line 182. The authors indicated “…in PE-induced cardiac hypertrophy treated……”, however SHR rats is a spontaneously Hypertensive Rats (SHR) model as stated by the authors in the Material and methods section.

-Figure 1A. What CK stand for? No indication is present in figures legend.
-Figure 2 “….. hypertrophy marker protein α-actinin (red)” should be “….. hypertrophy marker protein α-actinin (green)”
-Figure 4C. “miR-26a NC” should be “NC agomir”

- Lines 58/59. The authors stated “Hence, LC3-II/LC3-I ratio has been used to affect autophagy activity”; Actually, LC3-II/LC3-I ratio is used to measure autophagy levels and/or autophagy flux.

---

## Round 0.2 · Minor Revisions

The Referee has required to clarify whether the LC3 immunoblotting was quantified as LC3II/LC3I ratio or LC3II/GAPDH as requested.

Reviewer 1 ·

Basic reporting

no further comments

Experimental design

no further comments

Validity of the findings

no further comments

Additional comments

The authors should review the text and figures in order to replace the LC3II/LC3I ratio analysis with the LC3II/GAPDH ratio as requested.

Figure 1C - 1G – 6D: Did the authors analyze the levels of LC3II/GAPDH ratio? Do the new graphs refer to the measurement of the LC3II/GAPDH ratio? If so, the authors should modify the legend of the graphs in the figure and in the text that still report LC3II/LC3I ratio.

Manuscript Text that should be modified if LC3II/GAPDH results were reported:
Line 200, 203, 206, 207: LC3II/LC3I ratio should be corrected with LC3II/GAPDH or replaced by LC3II or lipidated LC3.
Line 210, 211: LC3 should be corrected with LC3II or lipidated LC3.
Line 216: LC3II/LC3I ratio should be corrected with LC3II/GAPDH.
Line 292: LC3II/LC3I ratio should be corrected with LC3II/GAPDH or replaced by LC3II or lipidated LC3.
Line 302: LC3II/LC3I ratio should be corrected with LC3II/GAPDH.
Line 335: LC3II/LC3I should be corrected with LC3II or lipidated LC3.
Figure 1F: Bar-A1 should be corrected with Baf-A1

---

## Round 0.3 · accepted · Accept

All concerns from the Reviewers were addressed.